# Variational Network Quantization

**Jan Achterhold[1,2], Jan M. Köhler[1], Anke Schmeink[2] & Tim Genewein[1,*]**

[1]Bosch Center for Artificial Intelligence
Robert Bosch GmbH
Renningen, Germany

[2]RWTH Aachen University
Institute for Theoretical Information Technology
Aachen, Germany
[*]Corresponding author: `tim.genewein@de.bosch.com`

## Abstract

In this paper, the preparation of a neural network for pruning and few-bit quantization is formulated as a variational inference problem. To this end, a *quantizing prior* that leads to a multi-modal, sparse posterior distribution over weights, is introduced and a differentiable Kullback-Leibler divergence approximation for this prior is derived. After training with Variational Network Quantization, weights can be replaced by deterministic quantization values with small to negligible loss of task accuracy (including pruning by setting weights to 0). The method does not require fine-tuning after quantization. Results are shown for ternary quantization on LeNet-5 (MNIST) and DenseNet (CIFAR-10).

## 1 Introduction

Parameters of a trained neural network commonly exhibit high degrees of redundancy (Denil et al., 2013) which implies an over-parametrization of the network. Network compression methods implicitly or explicitly aim at the systematic reduction of redundancy in neural network models while at the same time retaining a high level of task accuracy. Besides architectural approaches, such as SqueezeNet (Iandola et al., 2016) or MobileNets (Howard et al., 2017), many compression methods perform some form of *pruning* or *quantization*. Pruning is the removal of irrelevant units (weights, neurons or convolutional filters) (LeCun et al., 1990). Relevance of weights is often determined by the absolute value ("magnitude based pruning" (Han et al., 2016; 2017; Guo et al., 2016)), but more sophisticated methods have been known for decades, e.g., based on second-order derivatives (Optimal Brain Damage (LeCun et al., 1990) and Optimal Brain Surgeon (Hassibi & Stork, 1993)) or ARD (automatic relevance determination, a Bayesian framework for determining the relevance of weights, (MacKay, 1995; Neal, 1995; Karaletsos & Rätsch, 2015)). Quantization is the reduction of the bit-precision of weights, activations or even gradients, which is particularly desirable from a hardware perspective (Sze et al., 2017). Methods range from fixed bit-width computation (e.g., 12-bit fixed point) to aggressive quantization such as binarization of weights and activations (Courbariaux et al., 2016; Rastegari et al., 2016; Zhou et al., 2016; Hubara et al., 2016). Few-bit quantization (2 to 6 bits) is often performed by k-means clustering of trained weights with subsequent fine-tuning of the cluster centers (Han et al., 2016). Pruning and quantization methods have been shown to work well in conjunction (Han et al., 2016). In so-called "ternary" networks, weights can have one out of three possible values (negative, zero or positive) which also allows for simultaneous pruning and few-bit quantization (Li et al., 2016; Zhu et al., 2016).

This work is closely related to some recent Bayesian methods for network compression (Ullrich et al., 2017; Molchanov et al., 2017; Louizos et al., 2017; Neklyudov et al., 2017) that learn a posterior distribution over network weights under a sparsity-inducing prior. The posterior distribution over network parameters allows identifying redundancies through three means: weights with (1) an expected value very close to zero and (2) weights with a large variance can be pruned as they do not contribute much to the overall computation. (3) the posterior variance over non-pruned

parameters can be used to determine the required bit-precision (quantization noise can be made as large as implied by the posterior uncertainty). Additionally, Bayesian inference over model-parameters is known to automatically reduce parameter redundancy by penalizing overly complex models (MacKay, 2003).

In this paper we present *Variational Network Quantization* (VNQ), a Bayesian network compression method for simultaneous pruning and few-bit quantization of weights. We extend previous Bayesian pruning methods by introducing a multi-modal *quantizing prior* that penalizes weights of low variance unless they lie close to one of the target values for quantization. As a result, weights are either drawn to one of the quantization target values or they are assigned large variance values—see Fig. 1. After training, our method yields a Bayesian neural network with a multi-modal posterior over weights (typically with one mode fixed at $0$), which is the basis for subsequent pruning and quantization. Additionally, posterior uncertainties can also be interesting for network introspection and analysis, as well as for obtaining uncertainty estimates over network predictions (Gal & Ghahramani, 2015; Gal, 2016; Depeweg et al., 2016; 2017). After pruning and hard quantization, and without the need for additional fine-tuning, our method yields a deterministic feed-forward neural network with heavily quantized weights. Our method is applicable to pre-trained networks but can also be used for training from scratch. Target values for quantization can either be manually fixed or they can be learned during training. We demonstrate our method for the case of ternary quantization on LeNet-5 (MNIST) and DenseNet (CIFAR-10).

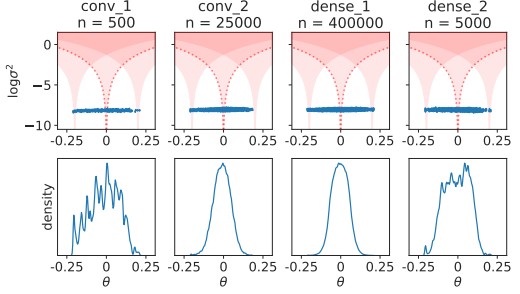 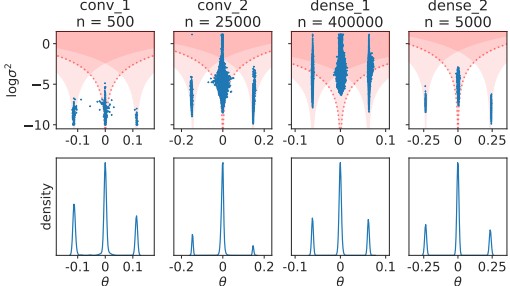

(a) Pre-trained network. No obvious clusters are visible in the network trained without VNQ. No regularization was used during pre-training.

(b) Soft-quantized network after VNQ training. Weights tightly cluster around the quantization target values.

Figure 1: Distribution of weights (means $\theta$ and log-variance $\log \sigma^2$) before and after VNQ training of LeNet-5 on MNIST (validation accuracy before: $99.2\%$ vs. after $195$ epochs: $99.3\%$). Top row: scatter plot of weights (blue dots) per layer. Means were initialized from pre-trained deterministic network, variances with $\log \sigma^2 = -8$. Bottom row: corresponding density[1]. Red shaded areas show the funnel-shaped "basins of attraction" induced by the quantizing prior. Positive and negative target values for ternary quantization have been learned per layer. After training, weights with small expected absolute value or large variance ($\log \alpha_{ij} \geq \log T_\alpha = 2$ corresponding to the funnel marked by the red dotted line) are pruned and remaining weights are quantized without loss in accuracy.

## 2 PRELIMINARIES

Our method extends recent work that uses a (variational) Bayesian objective for neural network pruning (Molchanov et al., 2017). In this section, we first motivate such an approach by discussing that the objectives of compression (in the minimum-description-length sense) and Bayesian inference are well-aligned. We then briefly review the core ingredients that are combined in Sparse Variational Dropout (Molchanov et al., 2017). The final idea (and also the starting point of our method) is to learn dropout noise levels per weight and prune weights with large dropout noise. Learning dropout noise per weight can be done by interpreting dropout training as variational inference of an approximate weight-posterior under a sparsity inducing prior - this is known as Variational Dropout which is described in more detail below, after a brief introduction to modern approximate posterior inference

---

[1] Kernel density estimate, with radial basis function kernels with a bandwidth of 0.05

in Bayesian neural networks by optimizing the evidence lower bound via stochastic gradient ascent and reparameterization tricks.

## 2.1 WHY BAYES FOR COMPRESSION?

Bayesian inference over model parameters automatically penalizes overly complex parametric models, leading to an *automatic regularization* effect (Grünwald, 2007; Graves, 2011) (see Molchanov et al. (2017), where the authors show that Sparse Variational Dropout (Sparse VD) successfully prevents a network from fitting unstructured data, that is a random labeling). The automatic regularization is based on the objective of maximizing model evidence, also know as marginal likelihood. A very complex model might have a particular parameter setting that achieves extremely good likelihood given the data, however, since the model evidence is obtained via marginalizing parameters, overly complex models are penalized for having many parameter settings with poor likelihood. This effect is also known as "Bayesian Occams Razor" in Bayesian model selection (MacKay, 2003; Genewein & Braun, 2014). The argument can be extended to variational Bayesian inference (with some caveats) via the equivalence of the variational Bayesian objective and the *Minimum description length* (MDL) principle (Rissanen, 1978; Grünwald, 2007; Graves, 2011; Louizos et al., 2017). The evidence lower bound (ELBO), which is maximized in variational inference, is composed of two terms: $\mathcal{L}^E$, the average message length required to transmit outputs (labels) to a receiver that knows the inputs and the posterior over model parameters and $\mathcal{L}^C$, the average message length to transmit the posterior parameters to a receiver that knows the prior over parameters:

$$\mathcal{L}^{\text{ELBO}} = \underbrace{\text{neg. reconstr. error}}_{-\mathcal{L}^E} + \underbrace{\text{neg. KL divergence}}_{-\mathcal{L}^C = \text{entropy} - \text{cross entropy}} .$$

Maximizing the ELBO minimizes the total message length: $\max \mathcal{L}^{\text{ELBO}} = \min \mathcal{L}^E + \mathcal{L}^C$, leading to an optimal trade-off between short description length of the data and the model (thus, minimizing the sum of error cost $\mathcal{L}^E$ and model complexity cost $\mathcal{L}^C$). Interestingly, MDL dictates the use of stochastic models since they are in general "more compressible" compared to deterministic models: high posterior uncertainty over parameters is rewarded by the entropy term in $\mathcal{L}^C$—higher uncertainty allows the quantization noise to be higher, thus, requiring lower bit-precision for a parameter. Variational Bayesian inference can also be formally related to the information-theoretic framework for lossy compression, rate-distortion theory, (Cover & Thomas, 2006; Tishby et al., 2000; Genewein et al., 2015). The only difference is that rate-distortion requires the use of *the optimal prior*, which is the marginal over posteriors (Hoffman & Johnson, 2016; Tomczak & Welling, 2017; Hoffman et al., 2017) - providing an interesting connection to empirical Bayes where the prior is learned from the data.

## 2.2 VARIATIONAL BAYES AND REPARAMETERIZATION

Let $\mathcal{D}$ be a dataset of $N$ pairs $(x_n, y_n)_{n=1}^N$ and $p(y|x, w)$ be a parameterized model that predicts outputs $y$ given inputs $x$ and parameters $w$. A *Bayesian neural network* models a (posterior) distribution over parameters $w$ instead of just a point-estimate. The posterior is given by Bayes' rule: $p(w|\mathcal{D}) = p(\mathcal{D}|w)p(w)/p(\mathcal{D})$, where $p(w)$ is the prior over parameters. Computation of the true posterior is in general intractable. Common approaches to approximate inference in neural networks are for instance: MCMC methods pioneered in (Neal, 1995) and later refined, e.g., via stochastic gradient Langevin dynamics (Welling & Teh, 2011), or variational approximations to the true posterior (Graves, 2011), Bayes by Backprop (Blundell et al., 2015), Expectation Backpropagation (Soudry et al., 2014), Probabilistic Backpropagation (Hernández-Lobato & Adams, 2015). In the latter methods the true posterior is approximated by a parameterized distribution $q_\phi(w)$. Variational parameters $\phi$ are optimized by minimizing the Kullback-Leibler (KL) divergence from the true to the approximate posterior $D_{\text{KL}}(q_\phi(w)||p(w|\mathcal{D}))$. Since computation of the true posterior is intractable, minimizing this KL divergence is approximately performed by maximizing the so-called "evidence

lower bound" (ELBO) or "negative variational free energy" (Kingma & Welling, 2014):

$$\mathcal{L}^{\text{ELBO}}(\phi) = \underbrace{\sum_{n=1}^{N} \mathbb{E}_{q_\phi(w)}[\log p(y_n|x_n, w)]}_{L_\mathcal{D}(\phi)} - D_{\text{KL}}(q_\phi(w)||p(w)), \tag{1}$$

$$\simeq \mathcal{L}^{\text{SGVB}}(\phi) = \frac{N}{M} \sum_{m=1}^{M} \log p(\tilde{y}_m|\tilde{x}_m, f(\phi, \epsilon_m)) - D_{\text{KL}}(q_\phi(w)||p(w)), \tag{2}$$

where we have used the *Reparameterization Trick*[2] (Kingma & Welling, 2014) in Eq. (2) to get an unbiased, differentiable, minibatch-based Monte Carlo estimator of the expected log likelihood $L_\mathcal{D}(\phi)$. A mini-batch of data is denoted by $(\tilde{x}_m, \tilde{y}_m)_{m=1}^{M}$. Additionally, and in line with similar work (Molchanov et al., 2017; Louizos et al., 2017; Neklyudov et al., 2017), we use the *Local Reparameterization Trick* (Kingma et al., 2015) to further reduce variance of the stochastic ELBO gradient estimator, which locally marginalizes weights at each layer and instead samples directly from the distribution over pre-activations (which can be computed analytically). See Appendix A.2 for more details on the Local reparameterization. Commonly, the prior $p(w)$ and the parametric form of the posterior $q_\phi(w)$ are chosen such that the KL divergence term can be computed analytically (e.g. a fully factorized Gaussian prior and posterior, known as the mean-field approximation). Due to the particular choice of prior in our work, a closed-form expression for the KL divergence cannot be obtained but instead we use a differentiable approximation (see Sec. 3.3).

## 2.3 VARIATIONAL INFERENCE VIA DROPOUT TRAINING

Dropout (Srivastava et al., 2014) is a method originally introduced for regularization of neural networks, where activations are stochastically dropped (i.e., set to zero) with a certain probability $p$ during training. It was shown that dropout, i.e., multiplicative noise on inputs, is equivalent to having noisy weights and vice versa (Wang & Manning, 2013; Kingma et al., 2015). Multiplicative Gaussian noise $\xi_{ij} \sim \mathcal{N}(1, \alpha = \frac{p}{1-p})$ on a weight $w_{ij}$ induces a Gaussian distribution

$$w_{ij} = \theta_{ij}\xi_{ij} = \theta_{ij}(1 + \sqrt{\alpha}\epsilon_{ij}) \sim \mathcal{N}(\theta_{ij}, \alpha\theta_{ij}^2) \tag{3}$$

with $\epsilon_{ij} \sim \mathcal{N}(0, 1)$. In standard (Gaussian) dropout training, the dropout rates $\alpha$ (or $p$ to be precise) are fixed and the expected log likelihood $L_\mathcal{D}(\phi)$ (first term in Eq. (1)) is maximized with respect to the means $\theta$. Kingma et al. (2015) show that Gaussian dropout training is mathematically equivalent to maximizing the ELBO (both terms in Eq. (1)), under a prior $p(w)$ and fixed $\alpha$ where the KL term does not depend on $\theta$:

$$\mathcal{L}(\alpha, \theta) = \mathbb{E}_{q_\alpha}[L_\mathcal{D}(\theta)] - D_{\text{KL}}(q_\alpha(w)||p(w)), \tag{4}$$

where the dependencies on $\alpha$ and $\theta$ of the terms in Eq. (1) have been made explicit. The only prior that meets this requirement is the scale invariant log-uniform prior:

$$p(\log|w_{ij}|) = \text{ const. } \Leftrightarrow p(|w_{ij}|) \propto \frac{1}{|w_{ij}|}. \tag{5}$$

Using this interpretation, it becomes straightforward to learn individual dropout-rates $\alpha_{ij}$ per weight, by including $\alpha_{ij}$ into the set of variational parameters $\phi = (\theta, \alpha)$. This procedure was introduced in (Kingma et al., 2015) under the name "Variational Dropout". With the choice of a log-uniform prior (Eq. (5)) and a factorized Gaussian approximate posterior $q_\phi(w_{ij}) = \mathcal{N}(\theta_{ij}, \alpha_{ij}\theta_{ij}^2)$ (Eq. (3)) the KL term in Eq. (1) is not analytically tractable, but the authors of Kingma et al. (2015) present an approximation

$$-D_{\text{KL}}(q_\phi(w_{ij})||p(w_{ij})) \approx \text{const.} + 0.5\log\alpha_{ij} + c_1\alpha_{ij} + c_2\alpha_{ij}^2 + c_3\alpha_{ij}^3, \tag{6}$$

see the original publication for numerical values of $c_1, c_2, c_3$. Note that due to the mean-field approximation, where the posterior over all weights factorizes into a product over individual weights $q_\phi(w) = \prod q_\phi(w_{ij})$, the KL divergence factorizes into a sum of individual KL divergences $D_{\text{KL}}(q_\phi(w)||p(w)) = \sum D_{\text{KL}}(q_\phi(w_{ij})||p(w_{ij}))$.

---

[2]The trick is to use a deterministic, differentiable (w.r.t. $\phi$) function $w = f(\phi, \epsilon)$ with $\epsilon \sim p(\epsilon)$, instead of directly using $q_\phi(w)$.

## 2.4 Pruning units with large dropout rates

Learning dropout rates is interesting for network compression since neurons or weights with very high dropout rates $p \to 1$ can very likely be pruned without loss in accuracy. However, as the authors of Sparse Variational Dropout (sparse VD) (Molchanov et al., 2017) report, the approximation in Eq. (6) is only accurate for $\alpha \leq 1$ (corresponding to $p \leq 0.5$). For this reason, the original variational dropout paper restricted $\alpha$ to values smaller or equal to 1, which are unsuitable for pruning. Molchanov et al. (2017) propose an improved approximation, which is very accurate on the full range of $\log \alpha$:

$$-D_{\mathrm{KL}}(q_\phi(w_{ij})||p(w_{ij})) \approx \mathrm{const.} + k_1 S(k_2 + k_3 \log \alpha_{ij}) - 0.5 \log(1 + \alpha_{ij}^{-1}) = F_{\mathrm{KL,LU}}(\theta_{ij}, \sigma_{ij}), \tag{7}$$

with $k_1 = 0.63576$, $k_2 = 1.87320$ and $k_3 = 1.48695$ and $S$ denoting the sigmoid function. Additionally, the authors propose to use an additive, instead of a multiplicative noise reparameterization, which significantly reduces variance in the gradient $\frac{\partial \mathcal{L}^{\mathrm{SGVB}}}{\partial \theta_{ij}}$ for large $\alpha_{ij}$. To achieve this, the multiplicative noise term is replaced by an exactly equivalent additive noise term $\sigma_{ij}\epsilon_{ij}$ with $\sigma_{ij}^2 = \alpha_{ij}\theta_{ij}^2$ and the set of variational parameters becomes $\phi = (\theta, \sigma)$:

$$w_{ij} = \theta_{ij} \underbrace{(1 + \sqrt{\alpha}\epsilon_{ij})}_{\mathrm{mult.noise}} = \theta_{ij} \underbrace{+\sigma_{ij}\epsilon_{ij}}_{\mathrm{add.noise}} \sim \mathcal{N}(\theta_{ij}, \sigma_{ij}^2), \quad \epsilon_{ij} \sim \mathcal{N}(0, 1). \tag{8}$$

After Sparse VD training, pruning is performed by thresholding $\alpha_{ij} = \frac{\sigma_{ij}^2}{\theta_{ij}^2}$. In Molchanov et al. (2017) a threshold of $\log \alpha = 3$ is used, which roughly corresponds to $p > 0.95$. Pruning weights that lie above a threshold of $T_\alpha$ leads to

$$\frac{\sigma_{ij}^2}{\theta_{ij}^2} \geq T_\alpha \Leftrightarrow \sigma_{ij}^2 \geq T_\alpha \theta_{ij}^2, \tag{9}$$

which means effectively that weights with large variance but also weights of lower variance and a mean $\theta_{ij}$ close to zero are pruned. A visualization of the pruning threshold can be seen in Fig. 1 (the "central funnel", i.e., the area marked by the red dotted lines for a threshold for $T_\alpha = 2$). Sparse VD training can be performed from random initialization or with pre-trained networks by initializing the means $\theta_{ij}$ accordingly. In Bayesian Compression (Louizos et al., 2017) and Structured Bayesian Pruning (Neklyudov et al., 2017), Sparse VD has been extended to include group-sparsity constraints, which allows for pruning of whole neurons or convolutional filters (via learning their corresponding dropout rates).

## 2.5 Sparsity inducing priors

For pruning weights based on their (learned) dropout rate, it is desirable to have high dropout rates for most weights. Perhaps surprisingly, Variational Dropout already implicitly introduces such a "high dropout rate constraint" via the implicit prior distribution over weights. The prior $p(w)$ can be used to induce sparsity into the posterior by having high density at zero and heavy tails. There is a well known family of such distributions: *scale-mixtures of normals* (Andrews & Mallows, 1974; Louizos et al., 2017; Ingraham & Marks, 2017):

$$w \sim \mathcal{N}(0, z^2); \quad z \sim p(z),$$

where the scales of $w$ are random variables. A well-known example is the spike-and-slab prior (Mitchell & Beauchamp, 1988), which has a delta-spike at zero and a slab over the real line. Gal & Ghahramani (2015); Kingma et al. (2015) show how Dropout training implies a spike-and-slab prior over weights. The log uniform prior used in Sparse VD (Eq. (5)) can also be derived as a marginalized scale-mixture of normals

$$p(w_{ij}) \propto \int \frac{1}{|z_{ij}|} \mathcal{N}(w_{ij}|0, z_{ij}^2) \mathrm{d}z_{ij} = \frac{1}{|w_{ij}|}; \quad p(z_{ij}) \propto \frac{1}{|z_{ij}|}, \tag{10}$$

also known as the normal-Jeffreys prior (Figueiredo, 2002). Louizos et al. (2017) discuss how the log-uniform prior can be seen as a continuous relaxation of the spike-and-slab prior and how the alternative formulation through the normal-Jeffreys distribution can be used to couple the scales of weights that belong together and thus, learn dropout rates for whole neurons or convolutional filters, which is the basis for Bayesian Compression (Louizos et al., 2017) and Structured Bayesian Pruning (Neklyudov et al., 2017).

## 3 VARIATIONAL NETWORK QUANTIZATION

We formulate the preparation of a neural network for a post-training quantization step as a variational inference problem. To this end, we introduce a multi-modal, quantizing prior and train by maximizing the ELBO (Eq. (2)) under a mean-field approximation of the posterior (i.e., a fully factorized Gaussian). The goal of our algorithm is to achieve *soft quantization*, that is learning a posterior distribution such that the accuracy-loss introduced by post-training quantization is small. Our variational posterior approximation and training procedure is similar to Kingma et al. (2015) and Molchanov et al. (2017) with the crucial difference of using a *quantizing prior* that drives weights towards the target values for quantization.

### 3.1 A QUANTIZING PRIOR

The log uniform prior (Eq. (5)) can be viewed as a continuous relaxation of the spike-and-slab prior with a spike at location $0$ (Louizos et al., 2017). We use this insight to formulate a quantizing prior, a continuous relaxation of a "multi-spike-and-slab" prior which has multiple spikes at locations $c_k$, $k \in \{1, \ldots, K\}$. Each spike location corresponds to one target value for subsequent quantization. The quantizing prior allows weights of low variance only at the locations of the quantization target values $c_k$. The effect of using such a quantizing prior during Variational Network Quantization is shown in Fig. 1. After training, most weights of low variance are distributed very closely around the quantization target values $c_k$ and can thus be replaced by the corresponding value without significant loss in accuracy. We typically fix one of the quantization targets to zero, e.g., $c_2 = 0$, which allows pruning weights. Additionally, weights with a large variance can also be pruned. Both kinds of pruning can be achieved with an $\alpha_{ij}$ threshold (see Eq. (9)) as in sparse Variational Dropout (Molchanov et al., 2017). Following the interpretation of the log uniform prior $p(w_{ij})$ as a marginal over the scale-hyperparameter $z_{ij}$, we extend Eq. (10) with a hyper-prior over locations

$$p(w_{ij}) = \int \mathcal{N}(w_{ij}|m_{ij}, z_{ij})p_z(z_{ij})p_m(m_{ij})\,\mathrm{d}z_{ij}\mathrm{d}m_{ij} \qquad p_m(m_{ij}) = \sum_k a_k \delta(m_{ij} - c_k), \quad (11)$$

with $p(z_{ij}) \propto |z_{ij}|^{-1}$. The location prior $p_m(m_{ij})$ is a mixture of weighted delta distributions located at the quantization values $c_k$. Marginalizing over $m$ yields the quantizing prior

$$p(w_{ij}) \propto \sum_k a_k \int \frac{1}{|z_{ij}|} \mathcal{N}(w_{ij}|c_k, z_{ij})\,\mathrm{d}z_{ij} = \sum_k a_k \frac{1}{|w_{ij} - c_k|}. \qquad (12)$$

In our experiments, we use $K = 3$, $a_k = 1/K$ $\forall k$ and $c_2 = 0$ unless indicated otherwise.

### 3.2 POST-TRAINING QUANTIZATION

Eq. (9) implies that using a threshold on $\alpha_{ij}$ as a pruning criterion is equivalent to pruning weights whose value does not differ significantly from zero:

$$\theta_{ij}^2 \leq \frac{\sigma_{ij}^2}{T_\alpha} \quad \Leftrightarrow \quad \theta_{ij} \in (-\frac{\sigma_{ij}}{\sqrt{T_\alpha}}, \frac{\sigma_{ij}}{\sqrt{T_\alpha}}). \qquad (13)$$

To be precise, $T_\alpha$ specifies the width of a scaled standard-deviation band $\pm\sigma_{ij}/\sqrt{T_\alpha}$ around the mean $\theta_{ij}$. If the value zero lies within this band, the weight is assigned the value $0$. For instance, a pruning threshold which implies $p \geq 0.95$ corresponds to a variance band of approximately $\sigma_{ij}/4$. An equivalent interpretation is that a weight is pruned if the likelihood for the value $0$ under the approximate posterior exceeds the threshold given by the standard-deviation band (Eq. (13)):

$$\mathcal{N}(0|\theta_{ij}, \sigma_{ij}^2) \geq \mathcal{N}(\theta_{ij} \pm \frac{\sigma_{ij}}{\sqrt{T_\alpha}}|\theta_{ij}, \sigma_{ij}^2) = \frac{1}{\sqrt{2\pi}\sigma_{ij}}e^{-\frac{1}{2T_\alpha}}. \qquad (14)$$

Extending this argument for pruning weights to a quantization setting, we design a post-training quantization scheme that assigns each weight the quantized value $c_k$ with the highest likelihood under the approximate posterior. Since variational posteriors over weights are Gaussian, this translates into minimizing the squared distance between the mean $\theta_{ij}$ and the quantized values $c_k$:

$$\arg \max_k \mathcal{N}(c_k|\theta_{ij}, \sigma_{ij}^2) = \arg \max_k e^{-\frac{(c_k - \theta_{ij})^2}{2\sigma_{ij}^2}} = \arg \min_k (c_k - \theta_{ij})^2. \qquad (15)$$

Additionally, the pruning rate can be increased by first assigning a hard $0$ to all weights that exceed the pruning threshold $T_\alpha$ (see Eq. (9)) before performing the assignment to quantization levels as described above.

### 3.3 KL DIVERGENCE APPROXIMATION

Under the quantizing prior (Eq. (12)) the KL divergence from the prior $D_{\text{KL}}(q_\phi(w)||p(w))$ to the mean-field posterior is analytically intractable. Similar to Kingma et al. (2015); Molchanov et al. (2017), we use a differentiable approximation $F_{\text{KL}}(\theta, \sigma, c)$[3], composed of a small number of differentiable functions to keep the computational effort low during training. We now present the approximation for a reference codebook $c = [-r, 0, r], r = 0.2$, however later we show how the approximation can be used for arbitrary ternary, symmetric codebooks as well. The basis of our approximation is the approximation $F_{\text{KL,LU}}$ introduced by Molchanov et al. (2017) for the KL divergence from a log uniform prior to a Gaussian posterior (see Eq. (7)) which is centered around zero. We observe that a weighted mixture of shifted versions of $F_{\text{KL,LU}}$ can be used to approximate the KL divergence for our multi-modal quantizing prior (Eq. (12)) (which is composed of shifted versions of the log uniform prior). In a nutshell, we shift one version of $F_{\text{KL}}$ to each codebook entry $c_k$ and then use $\theta$-dependent Gaussian windowing functions $\Omega(\theta)$ to mix the shifted approximations (see more details in the Appendix A.3). The approximation for the KL divergence from our multi-modal quantizing prior to a Gaussian posterior is given as

$$F_{\text{KL}}(\theta, \sigma, c) = \underbrace{\sum_{k:c_k \neq 0} \Omega(\theta - c_k) F_{\text{KL,LU}}(\theta - c_k, \sigma)}_{\text{local behavior}} + \underbrace{\Omega_0(\theta) F_{\text{KL,LU}}(\theta, \sigma)}_{\text{global behavior}} \quad (16)$$

with

$$\Omega(\theta) = \exp(-\frac{1}{2}\frac{\theta^2}{\tau^2}) \qquad \Omega_0(\theta) = 1 - \sum_{k:c_k \neq 0} \Omega(\theta - c_k). \quad (17)$$

We use $\tau = 0.075$ in our experiments. Illustrations of the approximation, including a comparison against the ground-truth computed via Monte Carlo sampling are shown in Fig. 2. Over the range of $\theta$- and $\sigma$-values relevant to our method, the maximum absolute deviation from the ground-truth is 1.07 nats. See Fig. 4 in the Appendix for a more detailed quantitative evaluation of our approximation.

This KL approximation in Eq. (16), developed for the reference codebook $c_r = [-r, 0, r]$, can be reused for any symmetric ternary codebook $c_a = [-a, 0, a], a \in \mathbb{R}^+$, since $c_a$ can be represented with the reference codebook and a positive scaling factor $s$, $c_a = sc_r$, $s = a/r$. As derived in the Appendix (A.4), this re-scaling translates into a multiplicative re-scaling of the variational parameters $\theta$ and $\sigma$. The KL divergence from a prior based on the codebook $c_a$ to the posterior $q_\phi(w)$ is thus given by $D_{KL}(q_\phi(w)||p_{c_a}(w)) \approx F_{\text{KL}}(\theta/s, \sigma/s, c_r)$. This result allows learning the quantization level $a$ during training as well.

## 4 EXPERIMENTS

In our experiments, we train with VNQ and then first prune via thresholding $\log \alpha_{ij} \geq \log T_\alpha = 2$. Remaining weights are then quantized by minimizing the squared distance to the quantization values $c_k$ (see Sec. 3.2). We use *warm-up* (Sønderby et al., 2016), that is, we multiply the KL divergence term (Eq. (2)) with a factor $\beta$, where $\beta = 0$ during the first few epochs and then linearly ramp up to $\beta = 1$. To improve stability of VNQ training, we ensure through clipping that $\log \sigma_{ij}^2 \in (-10, 1)$ and $\theta_{ij} \in (-a - 0.3679\sigma, a + 0.3679\sigma)$ (which corresponds to a shifted $\log \alpha$ threshold of 2, that is, we clip $\theta_{ij}$ if it lies left of the $-a$ funnel or right of the $+a$ funnel, compare Fig. 1). This leads to a clipping-boundary that depends on trainable parameters. To avoid weights getting stuck at these boundaries, we use gradient-stopping, that is, we apply the gradient to a so-called "shadow weight" and use the clipped weight-value only for the forward pass. Without this procedure our method still works, but accuracies are a bit worse, particularly on CIFAR-10. When learning codebook values

---

[3]To keep notation in this section simple, we drop the indices $ij$ from $w, \theta$ and $\sigma$ but we refer to individual weights and their posterior parameters throughout the section.

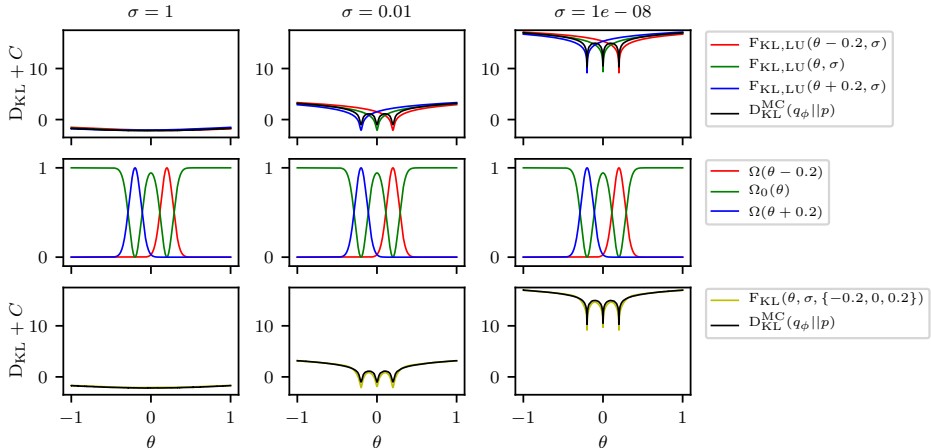

Figure 2: Approximation to the analytically intractable KL divergence $D_{KL}(q_\phi||p)$, constructed by shifting and mixing known approximations to the KL divergence from a log uniform prior to the posterior. Top row: Shifted versions of the known approximation (Eq. (7)) in color and the ground truth KL approximation (computed via Monte Carlo sampling) $D_{KL}^{MC}(q_\phi||p)$ in black. Middle row: weighting functions $\Omega(\theta)$ that mix the shifted known approximation to form the final approximation $F_{KL}$ shown in the bottom row (gold), compared against the ground-truth (MC sampled). Each column corresponds to a different value of $\sigma$. A comparison between ground-truth and our approximation over a large range of $\sigma$ and $\theta$ values is shown in the Appendix in Fig. 4. Note that since the priors are improper, KL approximation and ground-truth can only be compared up to an additive constant $C$ - the constant is irrelevant for network training but has been chosen in the plot such that ground-truth and approximation align for large values of $\theta$.

$a$ during training, we use a lower learning rate for adjusting the codebook, otherwise we observe a tendency for codebook values to collapse in early stages of training (a similar observation was made by Ullrich et al. (2017)). Additionally, we ensure $a \geq 0.05$ by clipping.

## 4.1 LeNet-5 on MNIST

We demonstrate our method with LeNet-5[4] (LeCun et al., 1998) on the MNIST handwritten digits dataset. Images are pre-processed by subtracting the mean and dividing by the standard-deviation over the training set. For the pre-trained network we run 5 epochs on a randomly initialized network (Glorot initialization, Adam optimizer), which leads to a validation accuracy of 99.2%. We initialize means $\theta$ with the pre-trained weights and variances with $\log \sigma^2 = -8$. The warm-up factor $\beta$ is linearly increased from 0 to 1 during the first 15 epochs. VNQ training runs for a total of 195 epochs with a batch-size of 128, the learning rate is linearly decreased from 0.001 to 0 and the learning rate for adjusting the codebook parameter $a$ uses a learning rate that is 100 times lower. We initialize with $a = 0.2$. Results are shown in Table 1, a visualization of the distribution over weights after VNQ training is shown in Fig. 1.

We find that VNQ training sufficiently prepares a network for pruning and quantization with negligible loss in accuracy and without requiring subsequent fine-tuning. Training from scratch yields a similar performance compared to initializing with a pre-trained network, with a slightly higher pruning rate. Compared to pruning methods that do not consider few-bit quantization in their objective, we achieve significantly lower pruning rates. This is an interesting observation since our method is based on a similar objective (e.g., compared to Sparse VD) but with the addition of forcing non-pruned weights to tightly cluster around the quantization levels. Few-bit quantization severely limits network capacity. Perhaps this capacity limitation must be countered by pruning fewer weights. Our pruning rates are roughly in line with other papers on ternary quantization, e.g., Zhu et al. (2016), who report sparsity levels between 30% and 50% with their ternary quantization method. Note that

---

[4]the Caffe version, see `https://github.com/BVLC/caffe/blob/master/examples/mnist/lenet_train_test.prototxt`

Table 1: Results on LeNet-5 (MNIST), showing validation error, percentage of non-pruned weights and bit-precision per parameter. Original is our pre-trained LeNet-5. We show results after VNQ training (without pruning and quantization, denoted by "no P&Q") where weights were deterministically replaced by the *full-precision means* $\theta$ and for VNQ training with subsequent pruning and quantization (denoted by "P&Q"). "random init." denotes training with random weight initialization (Glorot). We also show results of non-ternary or pruning-only methods (P): Deep Compression (Han et al., 2016), Soft weight-sharing (Ullrich et al., 2017), Sparse VD (Molchanov et al., 2017), Bayesian Compression (Louizos et al., 2017) and Stuctured Bayesian Pruning (Neklyudov et al., 2017).

| Method | val. error [%] | $\frac{|w \neq 0|}{|w|}$ [%] | bits |
|---|---|---|---|
| Original | 0.8 | 100 | 32 |
| VNQ (no P&Q) | 0.67 | 100 | 32 |
| VNQ + P&Q | 0.73 | 28.3 | 2 |
| VNQ + P&Q (random init.) | 0.73 | 17.7 | 2 |
| Deep Compression (P&Q) | 0.74 | 8 | $5-8$ |
| Soft weight-sharing (P&Q) | 0.97 | 0.5 | 3 |
| Sparse VD (P) | 0.75 | 0.7 | - |
| Bayesian Comp. (P&Q) | 1.0 | 0.6 | $7-18$ |
| Structured BP (P) | 0.86 | - | - |

a direct comparison between pruning, quantizing and ternarizing methods is difficult and depends on many factors such that a fair computation of the compression rate that does not implicitly favor certain methods is hardly possible within the scope of this paper. For instance, compression rates for pruning methods are typically reported under the assumption of a CSC storage format which would not fully account for the compression potential of a sparse ternary matrix. We thus choose not to report any measures for compression rates, however for the methods listed in Table 1, they can easily be found in the literature.

## 4.2 DENSENET ON CIFAR-10

Our second experiment uses a modern DenseNet (Huang et al., 2017) ($k = 12$, depth $L = 76$, with bottlenecks) on CIFAR-10 (Krizhevsky & Hinton, 2009). We follow the CIFAR-10 settings of Huang et al. (2017)[5]. The training procedure is identical to the procedure on MNIST with the following exceptions: we use a batch-size of 64 samples, the warm-up weight $\beta$ of the KL term is 0 for the first 5 epochs and is then linearly ramped up from 0 to 1 over the next 15 epochs, the learning rate of 0.005 is kept constant for the first 50 epochs and then linearly decreased to a value of 0.003 when training stops after 150 epochs. We pre-train a deterministic DenseNet (reaching validation accuracy of 93.19%) to initialize VNQ training. The codebook parameter for non-zero values $a$ is initialized with the maximum absolute value over pre-trained weights per layer. Results are shown in Table 2. A visualization of the distribution over weights after VNQ training is shown in the Appendix Fig. 3.

We generally observe lower levels of sparsity for DenseNet, compared to LeNet. This might be due to the fact that DenseNet already has an optimized architecture which removed a lot of redundant parameters from the start. In line with previous publications, we generally observed that the first and last layer of the network are most sensitive to pruning and quantization. However, in contrast to many other methods that do not quantize these layers (e.g., Zhu et al. (2016)), we find that after sufficient training, the complete network can be pruned and quantized with very little additional loss in accuracy (see Table 2). Inspecting the weight scatter-plot for the first and last layer (Appendix Fig. 3, top-left and bottom-right panel) it can be seen that some weights did not settle on one of the

---

[5]Our DenseNet($L = 76$, $k = 12$) consists of an initial convolutional layer ($3 \times 3$ with 16 output channels), followed by three dense blocks (each with 12 pairs of $1 \times 1$ convolution bottleneck followed by a $3 \times 3$ convolution, number of channels depends on growth-rate $k = 12$) and a final classification layer (global average pooling that feeds into a dense layer with softmax activation). In-between the dense blocks (but not after the last dense block) are (pooling) transition layers ($1 \times 1$ convolution followed by $2 \times 2$ average pooling with a stride of 2).

Table 2: Results on DenseNet (CIFAR-10), showing the error on the validation set, the percentage of non-pruned weights and the bit-precision per weight. Original denotes the pre-trained network. We show results after VNQ training without pruning and quantization (weights were deterministically replaced by the full-precision means $\theta$) denoted by "no P&Q", and VNQ with subsequent pruning and quantization denoted by "P&Q" (in the condition "(w/o 1)" we use full-precision means for the weights in the first layer and do not prune and quantize this layer).

| Method | val error [%] | $\frac{|w \neq 0|}{|w|}$ [%] | bits |
|---|---|---|---|
| Original | 6.81 | 100 | 32 |
| VNQ (no P&Q) | 8.32 | 100 | 32 |
| VNQ + P&Q (w/o 1) | 8.78 | 46 | 2 (32) |
| VNQ + P&Q | 8.83 | 46 | 2 |

prior modes (the "funnels") after VNQ training, particularly the first layer has a few such weights with very low variance. It is likely that quantizing these weights causes the additional loss in accuracy that we observe when quantizing the whole network. Without gradient stopping (i.e., applying gradients to a shadow weight at the trainable clipping boundary) we have observed that pruning and quantizing the first layer leads to a more pronounced drop in accuracy (about 3% compared to a network where the first layer is kept with full precision, not shown in results).

## 5 RELATED WORK

Our method is an extension of Sparse VD (Molchanov et al., 2017), originally used for network pruning. In contrast, we use a quantizing prior, leading to a multi-modal posterior suitable for few-bit quantization and pruning. Bayesian Compression and Structured Bayesian Pruning (Louizos et al., 2017; Neklyudov et al., 2017) extend Sparse VD to prune whole neurons or filters via group-sparsity constraints. Additionally, in Bayesian Compression the required bit-precision per layer is determined via the posterior variance. In contrast to our method, Bayesian Compression does not explicitly enforce clustering of weights during training and thus requires bit-widths in the range between 5 and 18 bits. Extending our method to include group-constraints for pruning is an interesting direction for future work. Another Bayesian method for simultaneous network quantization and pruning is soft weight-sharing (SWS) (Ullrich et al., 2017), which uses a Gaussian mixture model prior (and a KL term without trainable parameters such that the KL term reduces to the prior entropy). SWS acts like a probabilistic version of k-means clustering with the advantage of automatic collapse of unnecessary mixture components. Similar to learning the codebooks in our method, soft weight-sharing learns the prior from the data, a technique known as empirical Bayes. We cannot directly compare against soft weight-sharing since the authors do not report results on ternary networks. Gal et al. (2017) learn dropout rates by using a continuous relaxation of dropout's discrete masks (via the concrete distribution). The authors learn layer-wise dropout rates, which does not allow for dropout-rate-based pruning. We experimented with using the concrete distribution for learning codebooks for quantization with promising early results but so far we have observed lower pruning rates or lower accuracy compared to VNQ. A non-probabilistic state-of-the-art method for network ternarization is Trained Ternary Quantization (Zhu et al., 2016) which uses full-precision shadow weights during training, but quantized forward passes. Additionally it learns a (non-symmetric) scaling per layer for the non-zero quantization values, similar to our learned quantization level $a$. While the method achieves impressive accuracy, the sparsity and thus pruning rates are rather low (between 30% and 50% sparsity) and the first and last layer need to be kept with full precision.

## 6 DISCUSSION

A potential shortcoming of our method is the KL divergence approximation (Sec. 3.3). While the approximation is reasonably good on the relevant range of $\theta$- and $\sigma$-values, there is still room for improvement which could have the benefit that weights are drawn even more tightly onto the quantization levels, resulting in lower accuracy loss after quantization and pruning. Since our functional

approximation to the KL divergence only needs to be computed once and an arbitrary amount of ground-truth data can be produced, it should be possible to improve upon the approximation presented here at least by some brute-force function approximation, e.g., a neural network, polynomial or kernel regression. The main difficulty is that the resulting approximation must be differentiable and must not introduce significant computational overhead since the approximation is evaluated once for each network parameter in each gradient step. We have also experimented with a naive Monte-Carlo approximation of the KL divergence term. This has the disadvantage that local reparameterization (where pre-activations are sampled directly) can no longer be used, since weight samples are required for the MC approximation. To keep computational complexity comparable, we used a single sample for the MC approximation. In our LeNet-5 on MNIST experiment the MC approximation achieves comparable accuracy with higher pruning rates compared to our functional KL approximation. However, with DenseNet on CIFAR-10 and the MC approximation validation accuracy plunges catastrophically after pruning and quantization. See Sec. A.3 in the Appendix for more details. Compared to similar methods that only consider network pruning, our pruning rates are significantly lower. This does not seem to be a particular problem of our method since other papers on network ternarization report similar or even lower sparsity levels (Zhu et al. (2016) roughly achieve between 30% and 50% sparsity). The reason for this might be that heavily quantized networks have a much lower capacity compared to full-precision networks. This limited capacity might require that the network compensates by effectively using more weights such that the pruning rates become significantly lower. Similar trends have also been observed with binary networks, where drops in accuracy could be prevented by increasing the number of neurons (with binary weights) per layer. Principled experiments to test the trade-off between low bit-precision and sparsity rates would be an interesting direction for future work. One starting point could be to test our method with more quantization levels (e.g., $5, 7$ or $9$) and investigate how this affects the pruning rate.

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

## A    APPENDIX

### A.1    VISUALIZATION OF DENSENET WEIGHTS AFTER VNQ TRAINING

See Fig. 3.

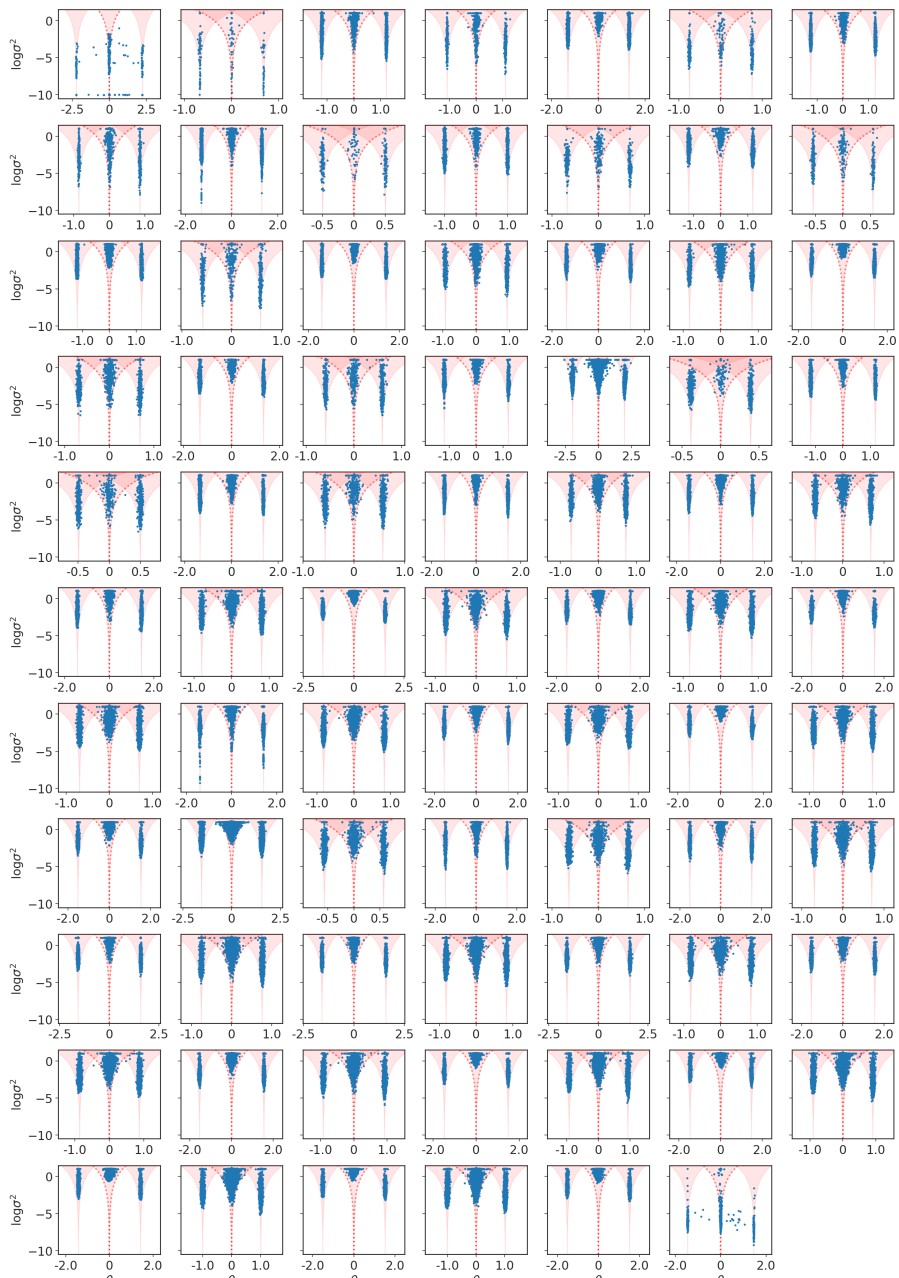

Figure 3: Visualization of distribution over DenseNet weights after training on CIFAR-10 with VNQ. Each panel shows one (convolutional or dense) layer, starting in the top-left corner with the input- and ending with the final layer in the bottom-right panel (going row-wise, that is first moving to the right as layers increase). The validation accuracy of the network shown is $91.68\%$ before pruning and quantization and $91.17\%$ after pruning and quantization.

## A.2 LOCAL REPARAMETERIZATION

We follow Sparse VD (Molchanov et al., 2017) and use the Local Reparameterization Trick (Kingma et al., 2015) and Additive Noise Reparmetrization to optimize the stochastic gradient variational lower bound $\mathcal{L}^{\text{SGVB}}$ (Eq. (2)). We optimize posterior means and log-variances $(\theta, \log \sigma^2)$ and the codebook level $a$. We apply Variational Network Quantization to fully connected and convolutional layers. Denoting inputs to a layer with $A^{M \times I}$, outputs of a layer with $B^{M \times O}$ and using local reparameterization we get:

$$b_{mj} \sim \mathcal{N}(\gamma_{mj}, \delta_{mj}); \;\; \gamma_{mj} = \sum_{i=1}^{I} a_{mi}\theta_{ij}, \;\; \delta_{mj} = \sum_{i=1}^{I} a_{mi}^2 \sigma_{ij}^2$$

for a fully connected layer. Similarly activations for a convolutional layer are computed as follows

$$\text{vec}(b_{mk}) \sim \mathcal{N}(\gamma_{mk}, \delta_{mk}); \;\; \gamma_{mk} = \text{vec}(A_m * \theta_k), \;\; \delta_{mk} = \text{diag}(\text{vec}(A_m^2 * \sigma_k^2)),$$

where $(\cdot)^2$ denotes an element-wise operation, $*$ is the convolution operation and $\text{vec}(\cdot)$ denotes reshaping of a matrix/tensor into a vector.

## A.3 KL APPROXIMATION FOR QUANTIZING PRIOR

Under the quantizing prior (Eq. (12)) the KL divergence from the log uniform prior to the mean-field posterior $D_{\text{KL}}(q_\phi(w_{ij})||p(w_{ij}))$ is analytically intractable. Molchanov et al. (2017) presented an approximation for the KL divergence under a (zero-centered) log uniform prior (Eq. (5)). Since our quantizing prior is essentially a composition of shifted log uniform priors, we construct a composition of the approximation given by Molchanov et al. (2017), shown in Eq. (7). The original approximation can be utilized to calculate a KL divergence approximation (up to an additive constant $\tilde{C}$) from a *shifted* log-uniform prior $p(w_{ij}) \propto \frac{1}{|w_{ij}-r|}$ to a Gaussian posterior $q_\phi(w_{ij})$ by transferring the shift to the posterior parameter $\theta$

$$D_{\text{KL}}\left(q_{\{\theta_{ij}, \sigma_{ij}\}}||p(w_{ij}) \propto \frac{1}{|w_{ij}-r|}\right) = D_{\text{KL}}\left(q_{\{\theta_{ij}-r, \sigma_{ij}\}}(w_{ij})||p(w_{ij}) \propto \frac{1}{|w_{ij}|}\right) + \tilde{C}, \;\; (18)$$

For small posterior variances $\sigma_{ij}^2$ ($\sigma_{ij} \ll r$) and means near the quantization levels (i.e., $|\theta_{ij}| \approx r$), the KL divergence is dominated by the mixture prior component located at the respective quantization level $r$. For these values of $\theta$ and $\sigma$, the KL divergence can be approximated by shifting the approximation $F_{\text{LU,KL}}(\theta, \sigma)$ to the quantization level $r$, i.e., $F_{\text{LU,KL}}(\theta \pm r, \sigma)$. For small $\sigma$ and values of $\theta$ near zero or far away from any quantization level, as well as for large values of $\sigma$ and arbitrary $\theta$, the KL divergence can be approximated by the original non-shifted approximation $F_{\text{LU,KL}}(\theta, \sigma)$. Based on these observations we construct our KL approximation by properly mixing shifted versions of $F_{\text{LU,KL}}(\theta \pm r, \sigma)$. We use Gaussian window functions $\Omega(\theta \pm r)$ to perform this weighting (to ensure differentiability). The remaining $\theta$ domain is covered by an approximation located at zero and weighted such that this approximation is dominant near zero and far away from the quantization levels, which is achieved by introducing the constraint that all window functions sum up to one on the full $\theta$ domain. See Fig. 2 for a visual representation of shifted approximations and their respective window functions.

### A.3.1 APPROXIMATION QUALITY

We evaluate the quality of our KL approximation (Eq. (16)) by comparing against a ground-truth Monte Carlo approximation on a dense grid over the full range of relevant $\theta$ and $\sigma$ values. Results of this comparison are shown in Fig. 4. Alternatively to the functional KL approximation, one could also use a naive Monte Carlo approximation directly. This has the disadvantage that local reparameterization can no longer be used, since actual samples of the weights must be drawn. To assess the quality of our functional KL approximation, we also compare against experiments where we use a naive MC approximation of the KL divergence term, where we only use a single sample for approximating the expectation to keep computational complexity comparable to our original method. Note that the "ground-truth" MC approximation used before to evaluate KL approximation quality uses many more samples which would be prohibitively expensive during training. To test for the effect of

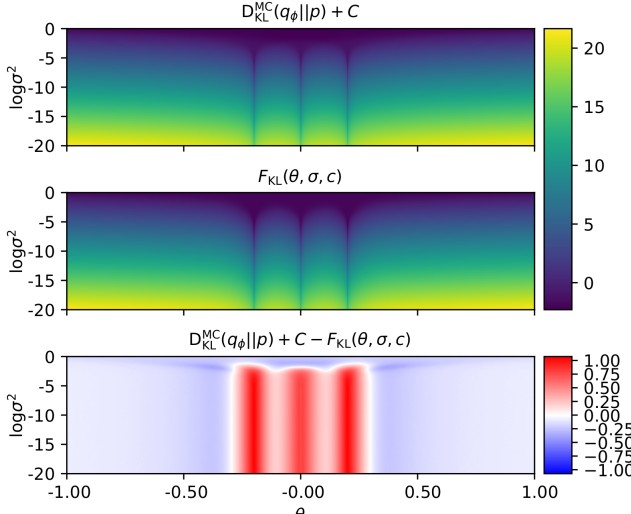

Figure 4: Quantitative analysis of the KL approxmiation quality. The top panel shows the "ground-truth" (computed via computationally expensive Monte Carlo approximation), the middle panel shows our approxiomation (Eq. (16)) and the bottom panel shows the difference between both. The maximum absolute error between our approximation and the ground-truth is 1.07 nats.

local reparameterization in isolation we also show results for our functional KL approximation without using local reparameterization. The results in Table 3 show that the naive MC approximation of the KL term leads to slightly lower validation error on MNIST (LeNet-5) (with higher pruning rates) but on CIFAR-10 (DenseNet) the validation error of the network trained with the naive MC approximation catastrophically increases after pruning and quantizing the network. Except for removing local reparameterization or plugging in the naive MC approximation, experiments were ran as described in Sec. 4.

Table 3: Comparing the effects of local reparameterization and naive MC approximation of the KL divergence. "func. KL approx" denotes our functional approximation of the KL divergence given by Eq. (16). "naive MC approx" denotes a naive Monte Carlo approximation that uses a single sample only. The first column of results shows the validation error after training, but without pruning and quantization (no P&Q), the next column shows results after pruning and quantization (results in brackets correspond to the validation error without pruning and quantizing the first layer).

| Setting | val. error no P&Q [%] | val. error P&Q [%] | $\frac{|w \neq 0|}{|w|}$ [%] |
|---|---|---|---|
| **LeNet-5 on MNIST** | | | |
| local reparam, func. KL approx | 0.67 | 0.73 | 28.3 |
| no local reparam, func. KL approx | 0.69 | 0.91 | 12.4 |
| no local reparam, naive MC approx | 0.6 | 0.69 | 8.8 |
| **DenseNet on CIFAR-10** | | | |
| local reparam, func. KL approx | 8.32 | 8.83 (8.78) | 46 |
| no local reparam, naive MC approx | 20.75 | 77.71 (75.74) | 60.7 |

Inspecting the distribution over weights after training with the naive MC approximation for the KL divergence, shown in Fig. 5 for LeNet-5 and in Fig. 6 for DenseNet, reveals that weight-means tend to be more dispersed and weight-variances tend to be generally lower than when training with our functional KL approximation (compare Fig. 1 for LeNet-5 and Fig. 3 for DenseNet). We speculate that the combined effects of missing local reparameterization and single-sample MC approximation lead to more noisy gradients.

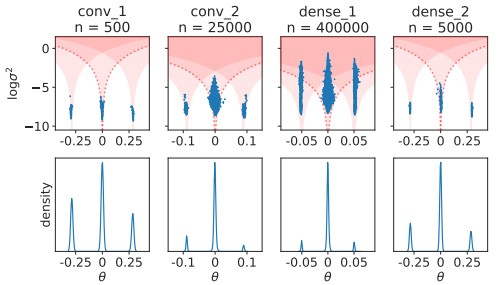

(a) No local reprametrization, functional KL approximation given by Eq. (16).

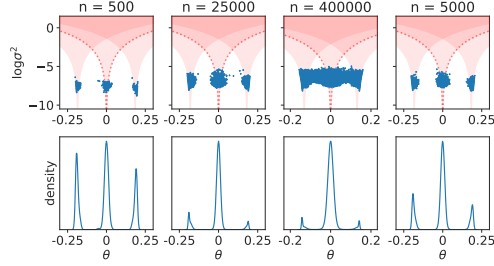

(b) No local reparameterization, naive MC approximation for KL divergence.

Figure 5: Distribution of weights after training without local reparameterization but with functional KL approximation (a) and after training with naive MC approximation (b). Top rows: scatter plot of weights (blue dots) per layer. Bottom row: corresponding density.

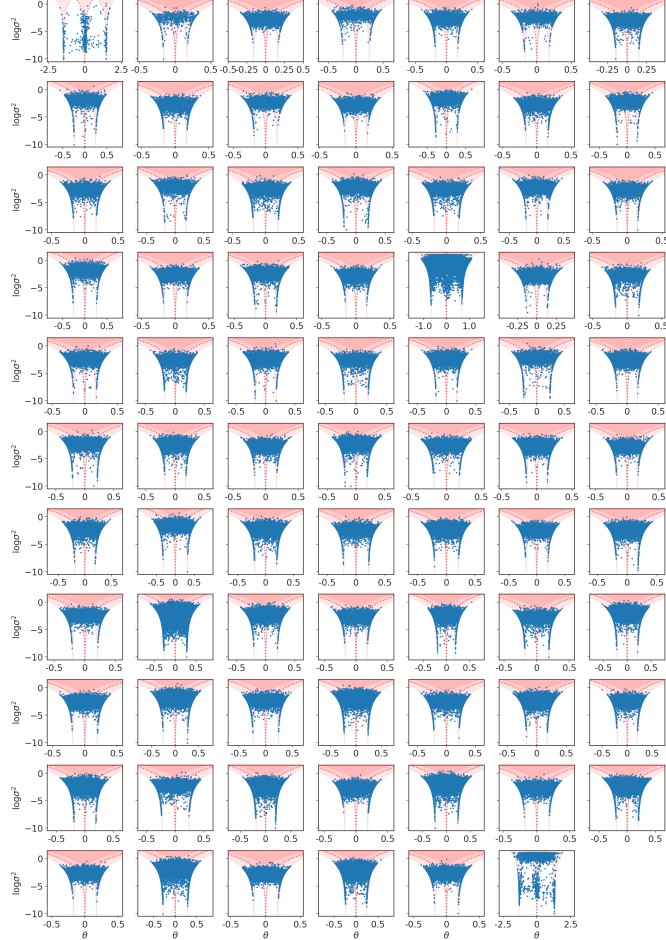

Figure 6: Visualization of distribution over DenseNet weights after training on CIFAR-10 with naive MC approximation for the KL divergence (and without local reparameterization). Each panel shows one layer, starting in the top-left corner with the input- and ending with the final layer in the bottom-right panel (going row-wise, that is first moving to the right as layers increase). Validation accuracy before pruning and quantization is 79.25% but plunges to 22.29% after pruning and quantization.

A.4 REUSING THE KL APPROXIMATION FOR ARBITRARY CODEBOOKS

We show that the KL approximation (Eq. (16)), developed for a fixed reference codebook, can be reused for arbitrary codebooks as long as codebook learning is restricted to learning a multiplicative scaling factor. Without loss of generality we consider the case of ternary, symmetric codebooks[6]

$$c_r = [-r, 0, r]; \quad p_{c_r}(w) = \sum_{k=1}^{3} \frac{a_k}{|w - c_{r,k}|} \tag{19}$$

where $r \in \mathbb{R}^+$ is the quantization level value and $p_{c_r}$ denotes a sparsity-inducing, quantizing prior over weights (sparsity is induced because one of the codebook entries is fixed to 0). We denote $c_r$ as the *reference* codebook for which we design the KL approximation $D_{\mathrm{KL}}(q_\phi(w)||p_{c_r}) = F_{\mathrm{KL}}(\theta, \sigma, c_r)$ (Eq. (16)). This approximation can be reused for any symmetric ternary codebook $c_a = [-a, 0, a]$ with quantization level $a \in \mathbb{R}^+$. The latter can be seen by representing $c_a$ with the reference codebook and a positive scaling factor $s > 0$ as $c_a = sc_r$, $s = a/r$. This re-scaling translates into a multiplicative re-scaling of the variational parameters $\theta$ and $\sigma$. To see this, consider the prior $p_{c_a}$, based on codebook $c_a$:

$$p_{c_a}(w) = \frac{1}{Z} \sum_{k=1}^{3} \frac{a_k}{|w - c_{a,k}|} = \frac{1}{Z} \sum_{k=1}^{3} \frac{a_k}{|w - sc_{r,k}|}. \tag{20}$$

The KL divergence from a prior based on the codebook $c_a$ to the posterior $q_\phi(w)$ is given by

$$
\begin{aligned}
D_{\mathrm{KL}}(q_\phi(w)||p_{c_a}(w)) &= \int q_\phi(w) \log \frac{q_\phi(w)}{\sum_{k=1}^{3} \frac{a_k}{|w - c_{a,k}|}} \, \mathrm{d}w + C \\
&= \int q_\phi(w) \log \frac{q_\phi(w)}{\frac{1}{s} \sum_{k=1}^{3} \frac{a_k}{|\frac{w}{s} - c_{r,k}|}} \, \mathrm{d}w + C \quad | \text{ subst. } z = \frac{w}{s}, \mathrm{d}w = s\mathrm{d}z \\
&= \int q_\phi(sz) \log \frac{q_\phi(sz)}{\frac{1}{s} \sum_{k=1}^{3} \frac{a_k}{|z - c_{r,k}|}} s\mathrm{d}z + C.
\end{aligned}
\tag{21}
$$

Since $q_\theta(sz)$ is Gaussian, the scaling $s$ can be transfered into the variational parameters $\phi = (\theta, \sigma)$:

$$q_\phi(sz) = \mathcal{N}(s; \theta, \sigma^2) = \frac{1}{s}\mathcal{N}(z; \frac{\theta}{s}, \frac{\sigma^2}{s^2}) = \frac{1}{s}q_{\hat\phi}(z),$$

with $\hat\phi = (\frac{\theta}{s}, \frac{\sigma}{s})$. Inserting into Eq. (21) yields:

$$
\begin{aligned}
D_{\mathrm{KL}}(q_\phi(w)||p_{c_a}(w)) &= \int \frac{1}{s} q_{\hat\phi}(z) \log \frac{\frac{1}{s}q_{\hat\phi}(z)}{\frac{1}{s} \sum_{k=1}^{3} \frac{a_k}{|z - c_{r,k}|}} s\mathrm{d}z + C. \\
&= \int q_{\hat\phi}(z) \log \frac{q_{\hat\phi}(z)}{\sum_{k=1}^{3} \frac{a_k}{|z - c_{r,k}|}} \, \mathrm{d}z + C. \\
&= D_{\mathrm{KL}}(q_{\hat\phi}(w)||p_{c_r}(w)) + C.
\end{aligned}
\tag{22}
$$

Thus, $D_{\mathrm{KL}}(q_\phi(w)||p_{c_a}(w)) = D_{\mathrm{KL}}(q_{\hat\phi}(w)||p_{c_r}(w)) + C \approx F_{\mathrm{KL}}(\theta/s, \sigma/s, c_r)$, where $F_{\mathrm{KL}}$ is given by Eq. (16). This means that the KL approximation can be used for arbitrary ternary, symmetric codebooks of the form $c_a = [-a, 0, a] = sc_r$ because the scaling factor $s$ translates into a re-scaling of the variational parameters $\hat\phi = (\frac{\theta}{s}, \frac{\sigma}{s})$.

---

[6]Note that indices $ij$ have been dropped for notational brevity from the whole section. However, throughout the section we refer to individual weights $w_{ij}$ and their variational parameters $\theta_{ij}$ and $\sigma_{ij}$

