# OpenReview forum: "Variational Network Quantization"
_ICLR.cc/2018/Conference — Accept (Poster)_

### Official Review · AnonReviewer2 · 2017-11-23
**Sparsity prior in variational Bayesian deep learning**

**Rating:** 7
**Confidence:** 5

**Review:**

This paper proposes to use a mixture of continuous spikes propto 1/abs(w_ij-c_k) as prior for a Bayesian neural network and demonstrates good performance with relatively sparsified convnets for minist and cifar-10. The paper is building quite a lot upon Kingma et al 2015 and  Molchanov et al 2017.

The paper is of good quality, clearly written with an ok level of originality and significance.

Pros:
1. Demonstrates a sparse Bayesian approach that scales.
2. Really a relevant research area for being able to make more efficient and compact deployment.
Cons:
1. Somewhat incremental relative to the papers mentioned above.
2. Could have taken the experimental part further. For example can we learn something about what part of the network has  the biggest potential for being pruned and use that to come up with modifications of the architecture?

---

> ### Author Response · Authors · 2018-01-05
> **Thanks for the review, short response to one of the issues raised**
>
>
> Regarding 2.: Connecting network compression with principled architecture search/optimization is a very interesting topic which has not received enough attention in the literature so far and the authors agree that there is promising potential. Unfortunately, our method might only be suitable for rather coarse statements. In order to provide interesting statements about parts of layers or even single neurons / convolutional filters, the method would need to be extended to include group-constraints as was done in Bayesian Compression or Structured Bayesian Pruning. This would allow statements about the relevance of certain sub-parts of networks. In contrast, our method only allows reporting sparsity-rates per layer, which could perhaps be used for high-level architecture exploration (layers with high sparsity can probably be made smaller).

---

### Official Review · AnonReviewer1 · 2017-11-27
**Good paper extending on previous work on variational compression for neural networks.**

**Rating:** 7
**Confidence:** 4

**Review:**

This paper presents Variational Network Quantization; a variational Bayesian approach for quantising neural network weights to ternary values post-training in a principled way. This is achieved by a straightforward extension of the scale mixture of Gaussians perspective of the log-uniform prior proposed at [1]. The authors posit a mixture of delta peaks hyperprior over the locations of the Gaussian distribution, where each peak can be seen as the specific target value for quantisation (including zero to induce sparsity). They then further propose an approximation for the KL-divergence, necessary for the variational objective, from this multimodal prior to a factorized Gaussian posterior by appropriately combining the approximation given at [2] for each of the modes. At test-time, the variational posterior for each weight is replaced by the target quantisation value that is closest, w.r.t. the squared distance, to the mean of the Gaussian variational posterior. Encouraging experimental results are shown with performance comparable to the state-of-the-art for ternary weight neural networks.

This paper presented a straightforward extension of the work done at [1, 2] for ternary networks through a multimodal quantising prior. It is generally well-written, with extensive preliminaries and clear equations. The visualizations also serve as a nice way to convey the behaviour of the proposed approach. The idea is interesting and well executed so I propose for acceptance. I only have a couple of minor questions:
- For the KL-divergence approximation you report a maximum difference of 1 nat per weight that seems a bit high; did you experiment with the `naive` Monte Carlo approximation of the bound (e.g. as done at Bayes By Backprop) during optimization? If yes, was there a big difference in performance?
- Was pre-training necessary to obtain the current results for MNIST? As far as I know, [1] and [2] did not need pre-training for the MNIST results (but did employ pre-training for CIFAR 10).
- How necessary was each one of the constraints during optimization (and what did they prevent)?
- Did you ever observe posterior means that do not settle at one of the prior modes but rather stay in between? Or did you ever had issues of the variance growing large enough, so that q(w) captures multiple modes of the prior (maybe the constraints prevent this)? How sensitive is the quantisation scheme?

Other minor comments / typos:
(1) 7th line of section 2.1 page 2, ‘a unstructured data’ -> ‘unstructured data’
(2) 5th line on page 3, remove ‘compare Eq. (1)’ (or rephrase it appropriately).
(3) Section 2.2, ’Kullback-Leibler divergence between the true and the approximate posterior’; between implies symmetry (and the KL isn’t symmetric) so I suggest to change it to e.g. ‘from the true to the approximate posterior’ to avoid confusion. Same for the first line of Section 3.3.
(4) Footnote 2, the distribution of the noise depends on the random variable so I would suggest to change it to a general \epsilon \sim p(\epsilon).
(5) Equation 4 is confusing.

[1] Louizos, Ullrich & Welling, Bayesian Compression for Deep Learning.
[2] Molchanov, Ashukha & Vetrov, Variational Dropout Sparsifies Deep Neural Networks.

---

> ### Author Response · Authors · 2018-01-05
> **Response to reviewer's questions/comments part I**
>
> We address the reveiewer's questions in their original order (due to limit in number of characters we respond with two separate entries)
>
> Did we try naive MC approximation of the bound?
> We ran additional experiments to compare our results against a naive MC approximation of the KL divergence. To keep computational complexity comparable to our method, we use a single sample for the MC approximation. On MNIST we get the same accuracy and even higher pruning rates, however on CIFAR-10 we get catastrophic accuracy after quantization and even the non-quantized network has significantly lower accuracy. We have added these results to the appendix A 3.1, including a new table and two figures.
>
> Was pre-training necessary on MNIST?
> We follow the same learning schedule as Sparse VD and train the first five epochs of a randomly initialized network without the KL penalization term and then gradually switch it on over the next epochs. We call the network after these first five epochs the "pre-trained" network, since five epochs suffice to get a decent MNIST classifier. We have run an additional experiment where we have a non-zero weight for the KL term already in the first epoch of training to start from a truly random network. Results were added to Table 1, training from scratch gets the same accuracy but slightly better pruning rates.
>
> How necessary was each of the constraints?
> Lower-bounding the log-variance helps avoiding numerical issues, upper-bounding the log-variance leads to higher accuracy during training - Bayesian Compression and the Multiplicative Normalizing Flows paper also report upper-bounding the posterior variance as it "helps avoiding bad local optima of the variational objective". Clipping the non-zero codebook levels at an absolute value of 0.05 to avoid getting collapsing codebooks was important since the objective implicitly favors close-to-zero codebook levels - particularly in the early stages of training such a collapse of the codebook needed to be prevented via clipping. Clipping weights that lie left to the left-most funnel or right to the right-most funnel helped with keeping accuracy after quantization. Without this clipping a small number of (seemingly important) weights are drawn to very large positive or negative values (particularly in the first layer). Since it is just a small number of weights, the impact on the objective is small, however quantizing such weights leads to significant accuracy loss. By clipping, the algorithm seems to find an alternative weight configuration that does not require such weights with large absolute values.
>
> Did we observe posterior means that do not settle at one of the prior modes?
> Yes, such cases can be seen in our experiments Fig. 1b (conv_1) and more pronounced in the first and last layer of DenseNet (top-left and bottom-right panel of Fig. 3 in the appendix). A small number of weights (blue dots) do not lie on the prior modes (outside the "funnels" in the low-variance regime). During early stages of training, the number of such weights is typically higher and quantizing such a network leads to poor accuracy. After sufficient training, we find in our experiments that a small number of such weights is tolerable without much loss in accuracy.

---

> > ### Author Response · Authors · 2018-01-05
> > **Response to reveiwer's questions/comments part II**
> >
> > Did we observe that the posterior variance of weights grows large enough to cover multiple prior modes?
> > Yes. Weights which are close to the upper \log \sigma clipping boundary (see Figure 1b and 3) have a comparatively large posterior variance such that all prior modes have a non-negligible likelihood. Empirically we find that this is not problematic for our method since such large variance weights are pruned after training (via thresholding \alpha, see Eq. 9 and the following sentence). A speculative explanation could be that these high-variance weights can essentially have arbitrary values since the information that they convey is discarded anyway downstream in the sparse network.
> >
> > How sensitive is the quantization scheme?
> > We found that training on MNIST typically worked quite robustly and was not severely affected by different initializations or changes in the learning rate etc. Training on CIFAR-10 was more sensitive regarding the learning rate. Probably the most crucial aspects were the clipping constraints and using a lower learning-rate for learning the codebook levels. One interesting aspect about the probabilistic soft-quantization is that weights with large posterior variance can essentially have any value that has sufficiently high likelihood under the posterior - this could be beneficial for improving robustness against hardware errors (rounding errors, limited precision, analog effects). In theory this should also translate into being more robust against noisy activations (or even network input) which could be very interesting. We think this question would require proper investigation beyond the scope of this paper.
> >
> >
> > Response to minor comments:
> > (1) Done.
> >
> > (2) Done.
> >
> > (3) Thanks for pointing it out, we have fixed this throughout the paper.
> >
> > (4) Done.
> >
> > (5) Another glitch, the equation should have been arranged differently (it should make more sense then) - we have updated the equation in the paper.

---

### Official Review · AnonReviewer3 · 2017-11-27
**A modern sparse Bayesian learning approach to weight quantization**

**Rating:** 7
**Confidence:** 3

**Review:**


The goal of this work is to infer weights of a neural network, constrained to a discrete set, where each weight can be represented by a few bits. This is a quite important and hot topic in deep learning. As a direct optimization would lead to a highly nontrivial combinatorial optimization problem, the authors propose a so-called 'quantizing prior' (actually a relaxed spike and slab prior to induce a sparsity enforcing heavy tail prior) over weights and derive a differentiable variational KL approximation. One important advantage of the current method is that this approach does not require fine-tuning after quantization. The paper presents ternary quantization for LeNet-5 (MNIST) and DenseNet-121 (CIFAR-10).

The paper is mostly well written and cites carefully the recent relevant literature. While there are a few glitches here and there in the writing, overall the paper is easy to follow. One exception is that in section 2, many ideas are presented in a sequence without providing any guidance where all this will lead.
The idea is closely related to sparse Bayesian learning but the variational approximation is achieved via the local reparametrization trick of Kingma 2015, with the key idea presented in section 3.3.



Minor

In the introduction, the authors write "... weights with a large variance can be pruned as they do not contribute much to the overall computation". What does this mean? Is this the marginal posterior variance as in ARD?

The authors write: "Additionally, variational Bayesian inference  is known to automatically reduce parameter redundancy by penalizing overly complex models." I would argue that
it is Bayesian inference; variational inference sometimes retains this property, but not always.

In Eq (10), z needs also subscripts, as otherwise the notation may suggest parameter tying. Alternatively, drop the indices entirely, as later in the paper.

Sec. 3.2. is not very well written. This seems to be the MAP of the product of the marginals,
or the mode of the variational distribution, not the true MAP configuration of the weight posterior. Please be more precise.

The abbreviation P&Q (probably Post-training Quantization) seems to be not defined in the paper.

---

> ### Author Response · Authors · 2018-01-05
> **Response to reviewer's comments**
>
> We address the reivewer's comments in the order in which they appear in the original review
>
>
> Section 2: no guidance where this will lead to - we added a short introduction to section 2 to tie the section together and provide an outline as a guidance to the reader. We also rewrote section 2.1 to be more focused.
>
> Minor comments:
>
> Intro: we write "... weights with a large variance can be pruned as they do not contribute much to the overall computation". What does this mean? Is this the marginal posterior variance as in ARD?
> Yes, in that sentence we refer to the marginal (approximate) posterior variance which is also the pruning criterion in ARD - however in ARD typically parameters with low variance (or high precision) are pruned. This is due to the fact that ARD assumes a zero-mean Gaussian prior over weights (with a different precision per parameter or group of parameters, that is adjusted during training and regularized by a hyper-prior). Weights that differ significantly from zero get assigned a high variance or, dually, weights with low variance are very likely to lie close to zero (the prior mean) and can thus be pruned. ARD is very similar to the situation where we only have the central funnel (a zero-mean prior) which is the case in Sparse Variational Dropout (compare Eq. 10 in our paper). However in the latter, as in our method, the pruning criterion takes into account both, the marginal posterior mean and variance (see Eq. 9) and also large-variance weights are pruned as long as the posterior mean is small enough (the intuition is that a high-variance weight can essentially have arbitrary values which implies that it most probably does not do anything sensible and can be pruned). To visualize the difference between the pruning criteria, consider the central funnel in the top-row plots of Figure 1: Sparse Variational Dropout and our method prune everything that lies within the area marked by the red dotted funnel. In contrast, thresholding the marginal posterior variance as in classical ARD would correspond to pruning everything that lies below a horizontal line in the "funnel plots" (which for the central funnel are precisely weights that lie close to zero). Note that of course different pruning criteria can also be used in ARD.
>
>
> Intro: Bayesian inference penalizes overly complex models, variational Bayesian inference does not necessarily do so - agreed, we have changed the sentence accordingly.
>
> Eq 10. - z needs subscripts - agreed, we have added sub-scripts throughout the paper.
>
> Section 3.2: do not refer to 'MAP' but be more precise - agreed, we rephrased our writing to refer to 'maximizing likelihood under the approximate posterior'.
>
> Clarify P&Q - P&Q refers to 'Pruning' and 'Quantization', we have clarified this in the corresponding table legends.

---

### Author Response · Authors · 2017-11-09
**Typo corrections**

The authors would like to correct four typos in the current version of the manuscript:
-) Table 1: Percentage of non-zero weights for Soft Weight-Sharing (P&Q) is 0.5 (not 3 as reported in the table) and bits for Deep Compression is 5 - 8 (not 10 - 13 as reported in the table)
-) Last paragraph before 4.1: We ensure alpha >= 0.05 by clipping.
-) Page 8, last sentence: we use a batch size of 64 samples
-) Appendix, Figure 3: The validation accuracy of the network shown is 91.55% (corresponds to VNQ (no P&Q) in Table 1).

We additionally want to clarify that in Eq. (11) p_m denotes the prior over locations whereas p_k is a scalar (the mixture weight for component k).

Note that the point of these corrections is to avoid potential confusion, our main results are not affected by these typos.

---

### Author Response · Authors · 2018-01-05
**Changelog for the updated revision of the paper**

We thank the reviewers for their feedback and constructive comments. Based on the feedback, we ran some additional experiments and made some changes to the paper. We have also re-ran our original experiments with two small modifications which produced slightly better results. We describe all changes below and respond to each reviewer individually with a separate comment on the corresponding review-entry in the forum.


Updated configuration for all experiments:

-) Changed pruning threshold to \log T_\alpha = 2 (was 3 in the first submission). Leads to small improvements in accuracy.

-) Gradient-stopping for clipping (applying gradients to a shadow weight at the clipping boundary that depends on the trainable codebook values and using the clipped weight only for the forward-pass). This helped improve results for CIFAR-10 experiment, particularly for quantizing the first layer without loss in accuracy. More details in Experiment section.



Additional experiments:

-) Performed experiments with naive MC approximation of KL divergence (single sample only) to compare against our functional approximation of the KL divergence. Good results on MNIST (same accuracy, higher pruning rates) but catastrophic results for quantized network on CIFAR-10 with the MC approximation. Results are shown in Appendix A 3.1.

-) Since the MC approximation cannot be used with local reparameterization, we performed a control experiment where we used our functional KL approximation but without local reparameterization (results in the Appendix, Table 3 and Figure 5a).



Paper changelog:

-) Updated results according to new experiment configuration (minor changes for MNIST in Table 1, better results for CIFAR-10 when quantizing the whole network shown in Table 2)

-) Added results for using a randomly initialized network (without any pre-training) on MNIST to Table 1. Same accuracy, slightly better pruning rates.

-) Added intro to section 2, to give some guidance to the reader

-) Rewrote 2.1 for more clarity.

-) No longer use the term 'MAP' but more accurately refer to 'maximizing likelihood under the approximate posterior' in 3.2.

-) Added the naive MC approximation of the KL divergence to the discussion.

-) Added detailed results of comparison between our KL approximation and the naive MC approximation to the appendix (A 3.1), including experiments on MNIST and CIFAR-10 (reported in Table 3) and two additional plots (Figure 5 and 6)

-) Changed abstract to use passive form

-) Fixed minor typos, glitches and other issues throughout the paper, including the ones pointed out by the reviewers.

---

### Author Response · Authors · 2018-02-22
**Changelog for camera-ready version**

Besides fixing typos, punctuation and brackets we clarify the precise DenseNet architecture used in our experiments:
following the nomenclature of the original DenseNet paper (Huang et al. 2016) we did not precisely use the DenseNet-121 architecture (as erroneously indicated in previous versions of the submission), but a DenseNet with a depth of 76 layers and a growth-rate of 12, i.e. DenseNet (L=76, k=12). Details regarding the network architecture were added in a footnote on page 9.

---

### Decision · Program_Chairs · 2018-01-29
**ICLR 2018 Conference Acceptance Decision**

**Decision:**

Accept (Poster)

**Comment:**

The paper presents a variational Bayesian approach for quantising neural network weights and makes interesting and useful steps in this increasingly popular area of deep learning.